# Augmented Reality for Human Decision Making and Human-Robot Collaboration: A Case Study in A Gasket Room in Manufacturing

1st Yuan Liu
*Australian Cobotics Centre*
*School of Architecture and Built Environment. Queensland University of Technology*
Brisbane, Australia
ORCID 0009-0005-5041-3289

*Abstract*—Human-Robot Collaboration (HRC) in manufacturing requires a balance between physical task execution and cognitive decision making. This study investigates the integration of Augmented Reality (AR) and robotics to enhance task performance in a real manufacturing setting. The research focuses on a gasket room, a critical part of the enclosure production process responsible for sealing enclosures. Initial observations identified a complex 24-step workflow involving physically demanding tasks, such as panel transportation and alignment, and cognitively demanding tasks including decision making. To enhance efficiency and quality, robots will be employed to handle repetitive and physically demanding processes, while humans will focus on the cognitively demanding process. AR will be used to design an HRC system where robots collaborate with humans in a real workspace, while AR will also be used to assist human decision making by providing real-time guidance. The proposed AR-based HRC system will be evaluated through user testing, measuring improvements in efficiency, accuracy, and cognitive workload. Despite the study's limitation of a small participant pool, future work will expand testing to a broader user group and explore scalability across different industrial settings.

*Keywords—augmented reality, human decision making, human-robot collaboration*

## I. INTRODUCTION

Human-Robot Collaboration (HRC) involves humans and robots working together on the same task within a shared space simultaneously [2]. In complex and dynamic tasks, robots excel at performing repetitive and physically demanding tasks, humans remain responsible for complex decision-making processes. This reliance on human decision making increases cognitive workload, which can lead to inefficiencies, errors, and decreased overall performance in collaborative tasks[7, 13]. Therefore, supporting human decision making is crucial to enhancing the collaboration between humans and robots, and improving task execution.

Our previous work indicates that non-robotic factors, such as task complexity, cognitive workload and user interface design have the most significant impact on human decision making during HRC tasks [11]. A promising solution to address these challenges is the adoption of immersive visualization technologies, such as Augmented Reality (AR). AR provides a low-cost, realistic, and immersive environment for HRC design [6]. It can also enhance HRC in multiple ways, including providing real-time visualization, instruction and guidance to improve safety [3, 8], and enable non-expert programming and development [9, 14]. Furthermore, AR provides interactive training environments [1, 10]and creates seamless user interfaces for human-robot interaction [12], making it a valuable tool in industrial settings.

Although previous studies have explored AR applications in industrial settings and human-robot collaboration, there is limited research on how AR can specifically support human decision making in real-time collaborative tasks. Most existing AR implementations focus on user interfaces [5], task and motion planning [6] or training [1] rather than dynamically assisting human workers in decision-making scenarios. This research aims to bridge this gap by investigating how AR can reduce cognitive workload and enhance decision efficiency in HRC environments.

The primary objectives of this study are:

1. To analyze task complexity and cognitive demands in a gasket room setting.

2. To design an AR-based HRC that supports human decision making during task execution.

3. To evaluate the proposed AR system in improving efficiency, accuracy, and user experience in HRC.

To achieve these objectives, a case study will be conducted in a gasket room of a manufacturing environment. The study will include data collection, data analysis, AR prototype development and evaluation to assess how AR can assist in decision-making processes. Data will be collected through observation and interviews. A combination of computer vision-based video analysis and thematic analysis of observation and interviews will be used to understand human behavior and decision-making processes during tasks. Based on that, an AR-based HRC prototype will be designed, then tested and refined with both experienced gasket room employees and non-expert users to ensure usability and adaptability across different skill levels.

This research aims to contribute to the field of human-robot collaboration by:

- Providing insights into best practices for AR-based designing HRC systems for real-world applications.

- Demonstrating how AR can enhance human decision making and reduce cognitive workload in industrial settings.

- Offering a framework for AR-supported decision making in collaborative robotics, which can be applied to other industrial environments.

By considering human decision making at the early stages of collaborative robot (cobot) adoption and system design, this research seeks to accelerate the integration of AR-assisted HRC solutions into industrial workflows.

## II. METHODOLOGY

This research follows a case study approach based on the qualitative method and human-centered design. It consists of four phases: data collection, data analysis, design and development, and evaluation (Fig.1).

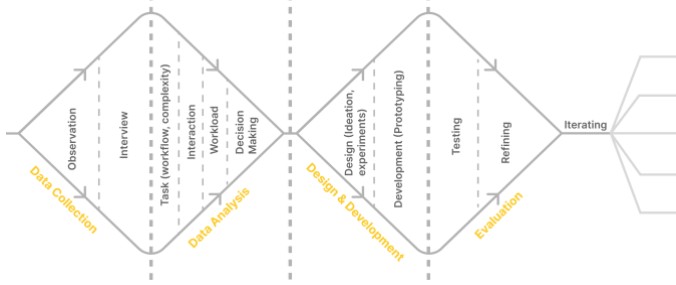

Fig. 1. Case Study Design, developed based on the Double Diamond Process Model [4].

### A. Data Collection

The data collection process includes observations and semi-structured interviews. Observations are conducted using 360-degree video recordings in a gasket room within manufacturing to capture real-world tasks and workplace dynamics. Two 360-degree cameras are placed at different locations to record continuously for 3 separate days, with each recording day occurring in a different week. The semi-structured interviews will be conducted with the same employees working in the gasket room to gain deeper insights into their experiences, challenges, and decision-making processes. A total of 3 employees will participate, with each interview lasting approximately 30 minutes. The study ensures ethical considerations by obtaining informed consent from all participants, maintaining confidentiality, and anonymizing collected data where necessary.

### B. Data Analysis

The video recordings will be analyzed using both a computer vision-based approach and thematic analysis, ensuring a comprehensive understanding of tasks, interaction, workload and human behavior, especially decision making. The computer vision analysis will employ techniques such as pose estimation,

and activity recognition to track human movement, assess task complexity, and identify physical demand, interaction, and decision-making patterns. Thematic analysis will be applied to both video observations and interview data to extract key themes related to task activities, interaction, physical and cognitive demand, decision making, and challenges in collaboration. For video analysis, key task sequences and worker interactions will be systematically coded based on predefined categories, such as task activities, human-human, human-object and human-machine interactions, physical and cognitive workload. The frame-by-frame analysis will help identify recurring patterns and challenges. These insights will be triangulated with interview data to provide a comprehensive understanding of the factors influencing humans during tasks.

The analysis will be conducted using OpenCV, Python-based libraries (e.g. media pipe), and ATLAS.ti. To ensure reliability, thematic coding will be performed by multiple researchers, with inter-rater agreement assessed. These combined methods will provide qualitative insights from both subjective and objective aspects, offering a deeper understanding of human decision making.

### C. Design and Development

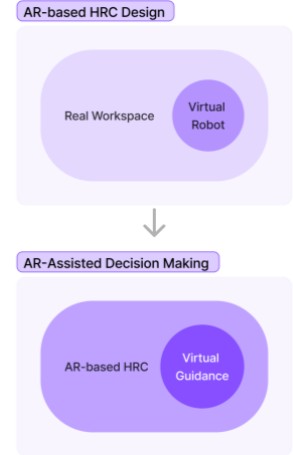

Fig. 2. Design and Development Process

Based on the results of the data analysis, an AR-based human-robot collaboration (HRC) system will be designed, and AR-assisted decision making will be implemented in this design (Fig. 2). The system will provide a virtual robot blended in the real workspace, real-time data visualization, task guidance, and feedback. A human-centered design approach will be employed, incorporating iterative prototyping and user feedback to ensure usability and effectiveness. The AR system will be developed in Unreal Engine 5 and implemented using Microsoft HoloLens and integrated with existing workflows.

### D. Evaluation

To evaluate the design, user testing will be conducted with both the same group of employees and non-expert groups to assess improvements in efficiency, accuracy, human-robot interaction quality, and the effectiveness of human decision-making. The evaluation will include both objective and subjective measures. Efficiency will be assessed based on task completion time, while accuracy will be measured through error

rates. Human-robot interaction quality will be evaluated using observational analysis and user feedback, and decision-making effectiveness will be assessed through structured tasks and self-reported confidence levels. The testing will take place in a real-world manufacturing environment, comparing AR-assisted HRC workflows with the original workflows.

## III. Preliminary Findings

Initial observations from September 2024 to October 2024 revealed that task workflow in the gasket room involves multiple breakdown steps and diverse requirements (Fig. 3, Fig. 4). The workflow consists of 24 distinct steps, each demanding varying levels of physical effort, cognitive processing—including decision making—and technological interaction. Physically demanding tasks include locating panels, transporting panels to the loading area, retrieving panels from the loading area, positioning panels on the shuttle table, aligning panels on the shuttle table etc., while cognitively demanding tasks involve measuring panel dimensions, comparing against computer values, preparing gasket machine, aligning panel on shuttle table, etc. These complexities can lead to increased cognitive workload, potential errors, and inefficiencies, ultimately impacting worker performance and safety.

Another challenge is ensuring seamless AR-based human-robot interaction in a dynamic manufacturing environment. Future work will focus on refining the AR design to provide more adaptive and context-aware guidance, improving real-time responsiveness to user needs. Additionally, integrating feedback from users will be essential to optimizing system functionality and enhancing worker acceptance.

Further analysis will explore how AR can specifically address these challenges to enhance worker performance and human-robot collaboration. The effectiveness of this approach will be evaluated through user testing, assessing improvements in task completion time, error rates, and worker cognitive load. Additionally, qualitative feedback from workers will be gathered to refine the system and ensure its usability in real-world manufacturing environments.

Future work will explore the scalability of this approach in different industrial settings beyond the gasket room. This includes evaluating how AR and robotics can be adapted to other industrial workflows.

## V. Conclusion

This study explored the integration of Augmented Reality

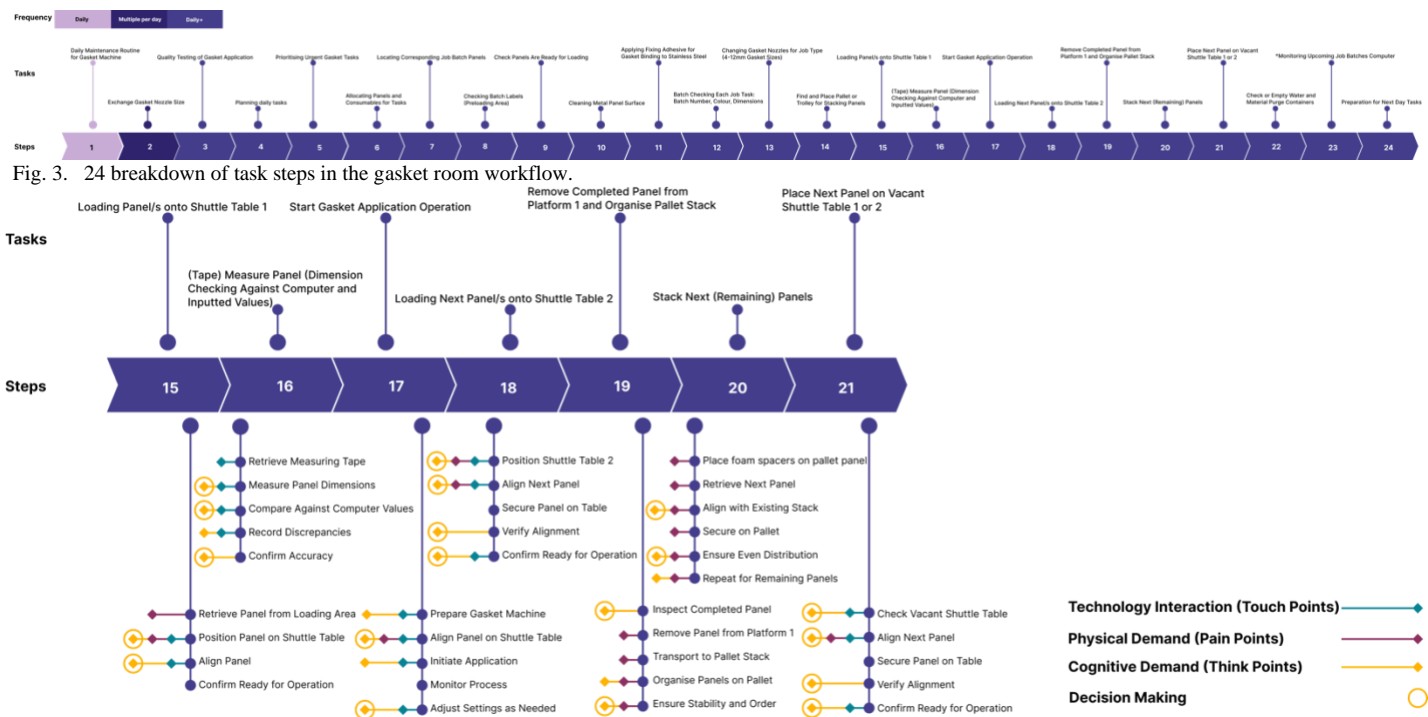

Fig. 3.   24 breakdown of task steps in the gasket room workflow.

Fig. 4.   Partial breakdown of task steps in the gasket room workflow.

(AR) and robotics to enhance human-robot collaboration (HRC)

## IV. Challenges and Future Work

One key challenge of this study is the limited number of employees working in the gasket room, which may affect the generalizability of the findings. To address this limitation, the design and evaluation process will be expanded to include a broader group of participants, incorporating user testing with non-experts. This will help assess the usability, adaptability, and learning curve of the AR-assisted system across different experience levels.

in a gasket room manufacturing environment. Initial observations revealed the 24 steps that require varying levels of physical effort, cognitive processing, and technological interaction. Physically demanding tasks, such as panel transportation and alignment, can be effectively delegated to robots, while AR has the potential to assist with cognitively demanding tasks by providing real-time guidance and visualization to enhance human decision-making process. Therefore, during the design and development phase, an AR-based HRC system will be developed, with AR also providing assistance for human decision-making.

ARC Industrial Transformation Training Centre (ITTC) for Collaborative Robotics in Advanced Manufacturing under Grant IC200100001.

Despite the study's limitations, particularly the restricted number of expert participants, further work will focus on expanding the evaluation to broader user groups. Refinements will include optimizing the AR-based HRC system based on testing results. Future studies will explore the system's scalability across different industrial settings

By leveraging AR and robotics, this research aims to understand and enhance human decision making in manufacturing environments, contributing to the broader adoption of human-robot collaboration systems.

## ACKNOWLEDGMENT

The author would like to acknowledge the support received through the following funding schemes of the Australian Government: ARC Industrial Transformation Training Centre (ITTC) for Collaborative Robotics in Advanced Manufacturing under Grant IC200100001.

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
