# OpenReview forum: "Augmented Reality for Human Decision Making and Human-Robot Collaboration: A Case Study in A Gasket Room in Manufacturing"
_humanrobotinteraction.org/HRI/2025/Workshop/VAM — HRI 2025 Workshop VAM Submission_

### Official Review · Reviewer_EGn4 · 2025-02-28

**Rating:** 7
**Confidence:** 5

**Review:**

The paper presents an interesting and timely idea to mitigate both physical and cognitive load using a blend of robotics and AR. The motivation however, could benefit from clearer articulations around the interactions between the HRC tasks and the role of AR in supporting such tasks. It was unclear from initial reading that there were two phases to this study (initial interviews to inform system design,  evaluation of designed system) --- it would benefit the paper to make this clear earlier. The descriptions of the analysis is clear, but it lacks grounding in relation to the task, i.e., a clear description of what the participants were asked to do is missing. Likewise, the description of the design and development of the system is quite sparse and makes it difficult to get a clear understanding of the work. Figure 1 is too small to discern  -- this should be enlarged and span both columns. The proposed directions and initial findings are promising that I think would benefit from substantial work future work (especially a study that includes a larger sample to confirm the findings presented in this paper).

---

### Decision · Program_Chairs · 2025-02-26

Accept